# Submicron-Sized Vermiculite Assisted Oregano Oil for Controlled Release and Long-Term Bacterial Inhibition

**DOI:** 10.3390/antibiotics10111324

**Published:** 2021-10-29

**Authors:** Sukitha Geethma Kothalawala, Jun Zhang, Yue Wang, Chengzhong Yu

**Affiliations:** 1Australian Institute of Bioengineering and Nanotechnology, University of Queensland, Brisbane, QLD 4072, Australia; s.kothalawala@uq.net.au (S.G.K.); yue.wang1@uq.edu.au (Y.W.); 2School of Chemistry and Molecular Engineering, East China Normal University, Shanghai 200241, China

**Keywords:** vermiculite, oregano essential oil, antibacterial, ball milling, volatility

## Abstract

Oregano essential oil (OEO) is a natural compound consisting of potent antibiotic molecules. Its volatility is the major obstacle against the transportation and anti-bacterial performance. In this work, submicron-sized vermiculite (SMV) particles were prepared from Australian vermiculite clay by ball milling, and tested as a potential particulate-carrier for OEO. The loading of OEO by SMV can be easily achieved by mechanical mixing. Compared to raw vermiculite and free OEO, the OEO-loaded SMV displayed sustained isothermal release behaviour of OEO and demonstrated enhanced antibacterial performance in in vitro antibacterial tests against *Escherichia coli* (*E. coli*) and *Staphylococcus* *epidermidis* (*S. epidermidis*). This study provides a facile and commercially viable approach in designing advantageous carriers for volatile actives in antimicrobial applications.

## 1. Introduction

In recent years, the food and agriculture industries have developed a growing interest in natural antibiotics with added benefits over synthetic antibiotics [1]. Issues concerning the safety of synthetic compounds have encouraged the production of more products from natural resources. OEO is a natural, essential oil with volatile secondary metabolites isolated from *Origanum vulgare* L. [2]. The antimicrobial and antioxidant properties of OEO are widely documented. Its active components include antioxidative agents such as carvacrol and thymol, and antibacterial agents such as monoterpene hydrocarbons (e.g., *p*-cymene and γ-terpinene) [3,4,5,6,7], altering bacterial cell permeability and causing leakage to induce bactericidal activity [8]. OEO as a phytogenic antimicrobial feed additive in poultry farming was reviewed by Alagwany and co-authors [9]. In the range of 300–1200 ppm, OEO exhibited effective antibacterial performance in livestock research, including poultry, cattle [10], hogs [11], and pet animals [12]. Furthermore, OEO contributes to multiple functions such as animal growth promotion [13,14], oxidation prevention [14], meat preservation [15], and digestion improvement [16,17]. However, the volatile nature of OEO has been a major limitation for transportation and field applications, leading to the rapidly decreasing effectiveness of OEO with the loss of active components over time.

To address this problem, various OEO encapsulation strategies have been developed to widen the applications of OEO as a food additive [18], antioxidant [19] and antibacterial agent [20]. These strategies mainly use emulsions/microemulsions [21] and chitosan-based materials [22]. It is noted that the release of chitosan-encapsulated OEO has been conducted in buffer solutions, while the OEO release was slowed down at neutral and alkaline conditions [21,23]. While this strategy is useful in developing pH-responsive release formulations, it does not address the volatility issue of OEO, which is an important parameter for a stable formulation in various application scenarios [24].

Natural vermiculite is one type of phyllosilicate, a clay mineral [25,26] available in Australia [27], among other regions, with diverse applications [28]. Vermiculite has a layered aluminosilicate structure [29], and the particle sizes of commercially available vermiculite are usually in the range of 1 mm to 1 cm [30]. By reducing the particle sizes, vermiculites have been utilized as soil amendments and fertilizer carriers [31]. This is because size reduction is advantageous in controlling the composition and increasing the specific surface area [32]. The size reduction of vermiculite can be accomplished via physical methods such as thermal shock [33,34,35], mechanical reduction [32,36], sonication [32,37,38,39] and chemical treatments, including ion exchange [40,41] or treatment with acid [42], H_2_O_2_ [43] and surfactants [44]. However, conventional techniques such as thermal shock are energy inefficient [45] due to the low thermal capacity of vermiculite. Sonication also suffers from high energy consumption, low conversion efficiency and low scalability for mass production [46]. While chemical techniques such as ion exchange [47,48] are promising, their applicability to scalable fabrication is yet to be demonstrated. Moreover, treatment by chemicals such as acids imposes a negative environmental impact [49]. It is important to develop a low-cost, scalable and green approach to prepare vermiculite with reduced sizes.

Ball milling is an efficient, fast and scalable mechanical size reduction method, which offers control over particle sizes in a broad range, such as submicron size [50,51]. Reducing the size into the submicron size range can dramatically increase the specific surface area, thus the performance can be significantly improved compared to raw materials. A higher specific surface area is beneficial in improving a number of properties, such as adsorption capacity, when used as a catalyst support or encapsulation agent for biomolecules [52]. In addition, chemical neutrality and biocompatibility [53] make vermiculite appealing in biological applications. Thus, submicron-sized vermiculite particles have been widely utilized for the delivery of antimicrobials such as chlorhexidine [54,55,56], metals [57] and metal oxide [58] nanoparticles. However, there are few reports on using submicron-sized vermiculite particles for OEO delivery and their antibacterial applications.

Herein, we report the successful fabrication of SMV particles as an advantageous carrier to enhance the thermal stability and antibacterial property of OEO. SMVs with particle size in the range of 200–1000 nm were produced by ball milling (Figure 1A). OEO can be loaded to SMV by mechanical mixing. In the isothermal release test, OEO-loaded SMV resulted in a slower release than free OEO at an elevated temperature of 60 °C, indicating improved retention of volatile OEO (Figure 1B). Furthermore, the OEO-loaded SMV shows excellent antimicrobial efficacy towards *E. coli* and *S. epidermidis* for 72 h (Figure 1C), surpassing the performance of OEO and OEO-loaded raw vermiculite.

## 2. Results

### 2.1. Characterization of the SMV

The study was conducted from the natural vermiculite collected in the state of Queensland, Australia. The collected vermiculite was dipped in HCl solution for carbonate removal. The resulting products were denoted as raw vermiculite (RV) after the process. Following the acid pre-treatment process, RV appeared as 1–5 mm-sized pieces in golden colour with low bulk density (Appendix A). After 30 min of pre-milling, the RV turned into a brown powder with an increased apparent density (data now shown). Scanning electron microscope (SEM) images of pre-milled vermiculite showed large particles with a wide size range of 2–50 μm (Appendix A). After 2 h of ball milling in the presence of 5 g water, SMV was fabricated with the appearance of fine brown powder (Appendix A). Dynamic laser scattering (DLS) measurements (Appendix A) suggested that the milled SMV particles were well dispersed in water. The average hydrodynamic diameter of the SMV was 459 nm with a polydispersity index of 0.44.

The morphology of SMV was directly observed by SEM and a transmission electron microscope (TEM). The low magnification SEM images of the SMV (Figure 2A) show submicron-sized particles in the range of 200–1000 nm, with irregular shapes. The magnified SEM image of the SMV (Figure 2B) shows that the particles possess a plate-like structure with thin thicknesses. The layered structures (indicated by the white arrows) of the vermiculite can be observed from the TEM images of the SMV (Figure 2C), and are consistent with the layered structure of vermiculite [59]. Energy-dispersive X-ray spectroscopy (EDS) analysis revealed the chemical composition of SMV. Al (yellow), Si (indigo) and O (red) elements were observed in the elemental mapping images (Figure 2D), in accordance with the aluminosilicate composition [60,61]. K (orange), Fe (purple) and Ti (light blue) Mg (green) elements were also observed in the elemental mapping, which were consistent with vermiculite’s molecular formula of [(Mg, Fe, Al, Ti)_3_[(AlSi)_4_O_10_]·(OH)_2_·4H_2_O [38,62]. Appendix A shows the elemental percentage of SMV quantified by EDS. O and Si hold the highest percentage (42.19% and 30.98%, respectively) in SMV. Meanwhile, Mg (6.68%) is more abundant than Al (1.61%), K (1.63%), Ti (0.24%) and Fe (1.79%).

In order to study the impact of ball milling on the crystallinity of vermiculite, the crystalline state of SMV was characterized by wide-angle X-ray diffraction (WA-XRD) compared to RV. The WA-XRD pattern of RV (Figure 3) showed a series of sharp peaks at 21, 29, 34, 39, 52 and 77°, which can be attributed to the 003, 004, 005, (20ℓ,13ℓ), (205, 134), (204, 135) and (138, 209) diffractions of crystalline vermiculite [38,63]. The narrow widths of these peaks indicate a large crystal domain size in the RV. Some characteristic peaks of RV crystals were not detected due to the orientation of bulk vermiculite chunks. In comparison, the WA-XRD pattern of SMV showed a series of notable peaks at 21, 23, 29–31, 36, 40, 43, 48, 52, 64 and 72°, which were attributed to the 003, (02) band, and the 004, 005, (20ℓ,13ℓ), 006, (205, 134), (204, 135), (206, 137) and (331, 060) diffractions, respectively. These characteristic peaks were significantly broadened, indicating a decreased crystalline domain size. 

### 2.2. Loading of Oregano Oil by SMV

SMV was used as a carrier to load OEO by mechanical mixing. Fourier-transform infrared spectroscopy (FTIR) was used to verify the successful OEO loading in the SMV. Figure 4 shows the FTIR spectrum of free OEO (green). Twenty obvious characteristic peaks were found at 638, 696 718, 751, 810, 863, 935, 993, 1054, 1116, 1172, 1243, 1297, 1356, 1423, 1459, 1507, 1583, 2869 and 2960 cm^−1^. The peaks in the range of 630–750 cm^−1^ can be attributed to the C-H bending of the aromatic ring of carvacrol, thymol and *p*-cymene [39]. The characteristic peaks in the range of 810–993 cm^−1^ can be attributed to the out-of-plane C-H wagging and the C-H bending from the 1:2:4-substitution of these active components [39]. The characteristic peaks at 1054, 1116 and 1172 cm^−1^ were attributed to para-substitution of *p*-cymene, ortho-substitution of carvacrol and meta-substitution of thymol, respectively [39]. The characteristic peaks at (1243, 1297 cm^−1^) and (1423, 1459 cm^−1^) can be attributed to the C-O stretching [64,65] and C=C aromatic vibration [36] of the active components. The characteristic peaks at 1356, 1507, 1583 were attributed to the asymmetric bending of the isopropyl group, C=C stretching carvacrol, thymol and *p*-cymene [39]. The characteristic peaks at 2869 and 2960 cm^−1^ can be attributed to the symmetric and asymmetric C-H stretching of the active components [64,66,67]. The spectrum of SMV (red) shows five characteristic peaks at 660, 724, 810, 1651 and 3717 cm^−1^, which can be attributed to FeO-OH vibrations [68], Si-O stretching [69], Mg-O stretching, the bending vibration of the inter-layer water and O-H stretching [70], respectively. Two broad bands in the range of 850–1200 and 3400 cm^−1^ were also observed in the spectra of SMV that are correlated to the -Si-O-Si and -Si-O-Al bonding [71] and O-H bending and deformation bands of molecular water [72,73,74], respectively. In the spectrum of OEO-loaded SMV, besides overlapping with the characteristic peaks of SMV, characteristic peaks at 1172, 1297, 1243, 1356, 1423, 1459, 1507, 1583, 1620, 2869 and 2960 cm^−1^ originated from OEO can still be observed, indicating the successful loading of OEO.

To quantitatively calculate the loading amount of OEO by SMV, a thermalgravimetric analysis (TGA) was conducted. As displayed in Figure 5A, free OEO shows a rapid weight loss of ~100% in the temperature range of 25–200 °C, indicating that free OEO evaporated completely before 200 °C, because most of the active components of OEO are volatile and have low boiling points < 201 °C [75]. The SMV showed 3.59% weight loss from the adsorbed moisture. The weight loss of OEO-loaded SMV was 8.67%. Hence, the loading amount of OEO is calculated to be 5.27%, which is in accordance with the feeding ratio of OEO and SMV, suggesting the complete loading of OEO by mechanical mixing.

Small-angle x-ray diffraction (SA-XRD) patterns were collected to investigate the layered structure of vermiculite before and after OEO loading. The SA-XRD pattern of RV showed two characteristic peaks at 2*θ* of 7.3° and 8.8°, corresponding to the gallery height of 1.40 and 1.16 nm, respectively [76]. Besides these characteristic peaks, the SA-XRD pattern of OEO-loaded RV also showed a shoulder peak at 2*θ* of 8.4°. This shoulder peak correlates to a wider *d* spacing of 1.22 nm, indicating that the adsorption of the organic components of OEO expanded the vermiculite layers. After ball milling, SMV only showed a broad and weak peak centred at 2*θ* of 8.5°, which indicated a *d* spacing of 1.21 nm. The broad diffraction peak of SMV was possibly attributed to the nonuniform layer spacing between vermiculite silicates and the mono- and double-hydrated Mg^2+^ forms after the size reduction by ball milling. The characteristic peak of *d* = 1.21 nm was unchanged after the loading of OEO in the SA-XRD pattern of OEO-loaded SMV, suggesting that the active components of OEO are mainly adsorbed on the surface of SMV, although the possibility of OEO located inside the layers could not be excluded.

### 2.3. Isothermal Release of Oregano Oil Loaded SMV

An elevated temperature of 60 °C was selected for the isothermal test since OEO is volatile and evaporates at high temperatures. The isothermal release profiles of free OEO, SMV and OEO-loaded SMV are shown in Figure 6. During the 14 h isothermal release process at 60 °C, 50% weight percentage of OEO was released within 4 h, and ~100% of OEO had completely evaporated after 11 h. The weight loss percentages of the SMV and OEO-loaded SMV were 3.61% and 5.79%, respectively. The weight loss of SMV came from the adsorbed moisture, consistent with TGA results. Based on the weight loss, it was calculated that 45.31% of the loaded OEO evaporated in the 14 h isothermal release at 60 °C. These data indicate that after loading by SMV, OEO shows a slower and sustained release.

### 2.4. In Vitro Antibacterial Tests

The antibacterial performance of various OEO formulations was tested in both gram-negative (*E. coli*) and gram-positive (*S. epidermidis*) models. The minimum inhibitory concentrations (MIC) of OEO and OEO-loaded SMV were obtained using the agar dilution method [77]. As shown in Figure 7, the free OEO treated group showed visible growth of bacterial colonies for both *E. coli* and *S. epidermidis* at OEO concentrations of 0.16–0.64 mg/mL. The inhibition to the bacterial colonies increased with the OEO concertation, and no visible bacteria growth was observed at an OEO concentration of 1.28 mg/mL, indicating the MIC value of OEO. The OEO-loaded SMV also showed dose-dependent inhibition of the growth of both *E. coli* and *S. epidermidis*. A much lower MIC value of 0.64 mg/mL was observed, indicating the OEO-loaded SMV exhibited efficient inhibition to bactericidal growth at this concentration. 

We further studied the long term antibacterial performance of the submicron-sized OEO formulation using a modified single plate serial dilution method [78]. According to Equation (1) in the Section 4.5.3., the colony forming units (CFU) per mL was calculated by counting the bacterial colonies with acceptable range of 6–60 CFU/mL under each dilution and choosing the lowest dilution with the correct number of colonies to multiply by the dilution factor. Even though MICs have been the gold standard to indicate the resistance of a microorganism to antimicrobial agents, these predictive values have been challenged by widespread antibiotic resistance. It appears the lag phase among the log, stationary, and death phases [79] of the typical bacterial growth curve in a culture medium initiates the adaptive bacterial mechanisms for new environments generated by antibiotics [80]. Consequently, an understanding of antibiotic effects on the lag phase is limited yet crucial. Therefore, three different concentrations of 0.64 mg/mL, 0.88 mg/mL, and 1.28 mg/mL of free OEO and OEO encapsulated SMV and RV formulations were selected to evaluate the response of the gram-negative and positive bacterial models during the lag phase. The growth of *E. coli* and *S. epidermidis* were evaluated for 72 h using the single plate dilution method at equal time-lapses of 24 h. 

The time-dependent bactericidal activities of OEO-loaded SMV, OEO-loaded RV, and free OEO were evaluated at OEO concentrations similar to the MIC (0.64, 0.88 and 1.28 mg/mL). Figure 8 shows that all groups exhibited dose-dependent bactericidal activity towards both *E. coli* and *S. epidermidis* at all time points between 24 and 72 h. At the lowest OEO concentrations of 0.64 mg/mL (CFU/mL 4.9 × 10^5^, 1.5 × 10^6^ and 9.2 × 10^6^ at 24, 48 and 72 h, respectively) and 0.88 mg/mL (CFU/mL 1.1 × 10^5^, 1.7 × 10^5^ and 9.3 × 10^5^ at 0.64 mg/mL at 24, 48 and 72 h, respectively), a gradual increase of viable *E. coli* colonies was observed for the free OEO group (Figure 8A). At the highest OEO concentration of 1.28 mg/mL, free OEO showed ≤1 × 10^2^ CFU/mL at 24 and 48 h and the regrowth of *E. coli* occurred at 72 h (CFU/mL 9.9 × 10^4^), indicating decay of bactericidal activity with time.

In comparison, the OEO-loaded SMV showed much higher bactericidal activity towards *E. coli* at all OEO concentrations. At the OEO concentration of 0.64 mg/mL, the OEO-loaded SMV showed limited viable colonies (CFU/mL < 1 × 10^2^) at 24 h and the regrowth of *E. coli* was observed at 48 and 72 h (CFU/mL 8 × 10^3^ and 1.4 × 10^4^). At a higher OEO concentration of 0.88 mg/mL, the OEO-loaded SMV showed limited viable colonies (CFU/mL < 1 × 10^2^) at 24 and 48 h and the regrowth of *E. coli* was only observed at 72 h (CFU/mL 6 × 10^3^). At the highest OEO concentration of 1.28 mg/mL, OEO-loaded SMV showed significant inhibition towards *E. coli* (CFU/mL < 1 × 10^2^) throughout 72 h, indicating long-term antibacterial activity. 

In order to investigate the contribution of the particle size of the OEO carrier, the long-term antibacterial efficacy of the loaded RV (size 1–5 mm) was also evaluated using the same method. At all three OEO concentrations, OEO-loaded RV showed more viable *E. coli* colonies than free OEO or OEO-loaded SMV (Figure 8A), indicating the lowest bactericidal activity. At the lowest OEO concentration of 0.64 mg/mL, the CFU/mL values of the OEO-loaded RV were 1.9 × 10^6^, 8.2 × 10^6^ and 1.2 × 10^8^ at 24, 48 and 72 h, respectively. At the OEO concentration of 0.88 mg/mL, the CFU/mL values of OEO-loaded RV were 8.7 × 10^5^, 2.1 × 10^5^ and 1.3 × 10^6^ at 24, 48 and 72 h, respectively. At the OEO concentration of 1.28 mg/mL, the CFU/mL values of OEO-loaded RV were 5.6 × 10^4^, 7.5 × 10^4^ and 1.5 × 10^5^ at 24, 48 and 72 h, respectively. The statistical analysis indicated a statistically significant difference among the 0.64 mg/mL groups (*p* = 0.0003 at 24 h and *p* < 0.0001 at 48 and 72 h). 

The in vitro antibacterial test was also carried out using the Gram-positive model, which showed a similar trend to the Gram-negative model (Figure 8B). At the lowest OEO concentrations of 0.64 mg/mL (CFU/mL 1.8 × 10^5^, 1.9 × 10^6^ and 1.2 × 10^7^ at 24, 48 and 72 h, respectively) and 0.88 mg/mL (CFU/mL 1.2 × 10^5^, 1.7 × 10^5^ and 1.3 × 10^6^ at 0.64 mg/mL at 24, 48 and 72 h, respectively), a gradual increase of viable *S. epidermidis* colonies was observed for the free OEO group (Figure 8A). At the highest OEO concentration of 1.28 mg/mL, free OEO showed ≤ 1 × 10^2^ CFU/mL at 24 h, and the regrowth of *S. epidermidis* occurred at 48 and 72 h (CFU/mL 1.2 × 10^4^ and 8.8 × 10^5^, respectively), indicating the decay of bactericidal activity towards *S. epidermidis* with time. The OEO-loaded SMV also showed much higher bactericidal activity towards *S. epidermidis* at all OEO concentrations. At almost all OEO concentrations and all time points, OEO-loaded SMV showed high antibacterial efficacy towards *S. epidermidis* with limited viable colonies (CFU/mL < 1 × 10^2^), showing long term bacterial inhibition. The regrowth of *S. epidermidis* only occurred at the OEO concentration of 0.64 mg/mL at 72 h (CFU/mL 3.8 × 10^4^). In comparison, OEO-loaded RV showed the lowest antibacterial activity among all three groups at all OEO concentrations. At the lowest OEO concentration of 0.64 mg/mL, the CFU/mL values of the OEO-loaded RV were 1.7 × 10^6^, 1.5 × 10^7^ and 2.0 × 10^8^ at 24, 48 and 72 h, respectively. At the OEO concentration of 0.88 mg/mL, the CFU/mL values of the OEO-loaded RV were 2.3 × 10^5^, 9.4 × 10^6^ and 1.4 × 10^7^ at 24, 48 and 72 h, respectively. At the OEO concentration of 1.28 mg/mL, the CFU/mL values of the OEO-loaded RV were 8.4 × 10^4^, 1.9 × 10^5^ and 1.5 × 10^5^ at 24, 48 and 72 h, respectively. The statistical analysis indicated a statistically significant difference among the 0.64 mg/mL groups of OEO and RV plus OEO at all time points (*p* < 0.0001). 

In order to substantiate the contribution to bacterial inhibition from the carrier, a long-term antibacterial test for SMV and RV without OEO was conducted. The results showing the growth control of the bacteria (GC, untreated group) and the background control (BC) are summarized in Appendix A. The carrier concentration was kept the same, with the OEO-loaded SMV and OEO-loaded RV groups having an OEO concentration of 0.88 mg/mL. Compared to the untreated groups, both SMV and RV showed negligible inhibition of bacteria growth for both *E. coli* and *S. epidermidis*, indicating the bacteria inhibition contribution from the carriers is limited. 

To investigate the action mechanism of bacterial inhibition, SEM was employed to visualize the morphological changes of the bacteria upon exposure to the OEO-loaded SMV at the OEO concentration of 0.88 mg/mL for 1 h. The untreated *S. epidermidis* showed a spherical morphology with an intact and relatively smooth surface (Figure 9a). The untreated *E. coli* showed a rod shape, and the bacterial wall remained intact (Figure 9b). After being treated by free OEO and OEO-loaded SMV, disordered wrinkling was observed on the surface of *S. epidermidis* (Figure 9c,e), indicating partial membrane disintegration and deformation. SMV was observed to adhere to the surface of *S. epidermidis* (indicated by arrows). Similarly, the free OEO treated *E. coli* showed wrinkled bacterial walls (Figure 9d). By comparison, the OEO-loaded SMV treated *E. coli* still maintained rod morphology but showed obvious bacterial wall damage (Figure 9f). It appears that OEO interacts with and dissolves the outer membranes of both the gram-positive and gram-negative strains, making the bacteria more likely to die. 

## 3. Discussion

This study used a top-down ball milling approach to successfully convert bulk vermiculite into submicron-sized clay particles. Australian raw vermiculite was selected as a cheap and abundant material source. As indicated by the DLS (Appendix A) and TEM (Figure 2) measurements, the milled SMV particles had an average size of 459 nm with a PDI of 0.44, indicating the particles had an overall submicron size but were not very uniform [81]. Apparently, the ball milling approach significantly reduced the particle size, thickness and crystalline domain sizes of vermiculite, as evidenced from the WA-XRD studies (Figure 3). Consequently, an increased specific surface area of SMV due to the size reduction was expected compared to that of RV, which is beneficial for loading of the active component of OEO and possibly other cargo molecules.

As an essential oil, the volatility of OEO hinders its application as an antibacterial additive [82]. It is noted that natural vermiculites with thermal stability and chemical inertness have been used to encapsulate and stabilize essential oils such as lemon oil [83]. Therefore, in our study, SMV was selected as a carrier of OEO to overcome the challenge of its instability. The loading of OEO by SMV was achieved simply by mechanical mixing, as demonstrated by the FTIR and TGA analyses. The location of the OEO in RV and SMV after loading was not very clear. It is possible that OEO was adsorbed at least partially in the layers of RV. It is deduced that the primary interactions between OEO and the SMV were from the Van de Waals interaction and hydrogen bonding [84]. A similar mechanism was proposed in the case of raw vermiculite loaded with lemon essential oil [83]. Because vermiculite has been used to load not only essential oils but also many other drug compounds, [54,85,86] it is expected that SMV may be used a platform to load a variety of active molecules.

The stability of essential oils is essential for their application performance [87], among which thermal stability is vital [87], especially for OEO, which is highly volatile [88]. From the isothermal release test of the OEO, it was shown that SMV had the advantage of slowing down the OEO release and significantly improving thermal stability (Figure 6). Because vermiculite [89] and OEO [90] are both insoluble in water, the sequential dilution method is prone to generate complications in MIC tests for accurate optical density measurements due to precipitation and oil separation. Therefore, the agar dilution method was utilized in our study [77,91,92]. The results (Figure 7) showed that the OEO-loaded SMV formulation exhibited better bacterial growth inhibition than the OEO on Gram-positive and Gram-negative bacterial strains. Considering the fact that vermiculite is chemically neutral and non-toxic, [93] the improved performance should be mainly attributed to the enhanced stability of OEO and the larger number of contact sites between the OEO and the bacteria. 

The advantage of SMV over RV is mainly reflected in their long-term antibacterial performance. Figure 8 shows that OEO-loaded SMV possessed long-term bactericidal activity against both Gram-positive and Gram-negative bacterial strains at relatively high OEO concentrations, surpassing those of the free OEO and the OEO-loaded RV. The volatility of free OEO is considered an important factor contributing to its shorter antimicrobial duration and lower killing effect compared to OEO-loaded SMV. On the other hand, both free OEO and RV (expended vermiculite) have low density or apparent density [53] and may float on the bacterial culture medium, leading to limited transportation of OEO to bacteria. In comparison, the apparent density of SMV is highly increased after the ball milling process. As both SMV and RV show neglectable killing effects on bacteria (Appendix A), we surmised the OEO/particle suspension contacts bacteria through sedimentation [94], contributing to higher OEO transportation and bacteria-killing effects through bacterial membrane/wall damage. Furthermore, the low surface-area-to-volume ratio of RV compared to SMV negatively contributes to the bactericidal activity, as the large surface area of the particles enhances their interaction with the microbes, facilitating their ability to carry out broad-spectrum antimicrobial activities [95].

The bactericidal mechanisms of OEO-loaded SMV were investigated by observing treated bacterial morphology, which indicated SMV adhesion and bacterial membrane/wall disintegration. The permeability of the membrane was attributed to the active ingredients of OEO, including carvacrol [96,97,98] and thymol [99,100], which are the major antibacterial components in OEO. OEO released to the media from the SMV surface weakened the bacterial membrane by increasing its permeability. It is possible that small-sized particles cause synergetic chemical and physical damage to bacterial membranes [101].

## 4. Materials and Methods

### 4.1. Chemicals and Reagents

The grade 3 vermiculite used in the present study was purchased from Queensland, Australia. Oxoid^TM^ Tryptone was purchased from Thermo-Fisher Scientific (Waltham, MA, USA). Yeast extract (granulated), NaCl (≥99.5%), agar (≤5% impurities) and HCl (37% solution) were purchased from Sigma Aldrich (St Louis, MO, USA). Oregano oil was purchased from Chem-Supply Pty Ltd. (Gillman, Australia). Deionized water (Millipore 18 mΩ/cm water solution) was provided from the University of Queensland chemical store and was used to prepare all solutions/dispersions. All the other reagents were of analytical reagent grade.

### 4.2. Synthesis of SMV

SMV was prepared by a ball milling method. Before the milling process, the grade 3 Australian vermiculite was pre-treated with HCl to dissolve possible carbonates. ~100 g of the grade 3 Australian vermiculite was weighed and soaked in 2 L of 0.1 mM HCl solution for 5 min. The treated vermiculite was filtered, washed three times with deionized water, and dried overnight in a 50 °C vacuum oven. The pre-treated sample was denoted as RV. As the RV was expanded vermiculite with very low apparent density, a 30-min pre-milling step was carried out to reduce the apparent density of the RV. In this pre-milling step, 10 g of RV and 5–10 mm agate balls were placed in a 250 mL agate grinding bowl, and the mixture was milled in a Fritsch^®^ Planetary Mill PULVERISETTE 5 classic line at a speed of 300 rpm for 30 min. Sequentially, 90 g of pre-milled vermiculite, 5 g of water, and the 5–10 mm agate balls were placed in the agate bowl and milled at 300 rpm for 2 h. The product obtained was denoted as SMV.

### 4.3. Characterizations

The morphology of the SMV was observed using a JEOL JSM 7800 SEM operated at 5 kV. For SEM observation, the samples were prepared by attaching the powder samples to the conductive carbon film on the SEM mounts. The TEM images were obtained using a JEOL 2100 microscope operated at 100 kV. The TEM specimens were prepared by dispersing the samples in ethanol after ultrasonication for 5 min and then depositing them directly onto a carbon film-supported copper grid. Energy-dispersive X-ray spectroscopy elemental mappings were conducted in the high angle annular dark-field (HAADF) scanning transmission electron microscopy (STEM) mode. Particle size distribution was measured from the DLS of a Zetasizer Nano-ZS from Malvern Instruments.

### 4.4. OEO Loading and Isothermal Release

The OEO was loaded by SMV or RV by mechanical mixing with the OEO:vermiclulite ratio of 5:95. The FTIR spectra of OEO-loaded SMV and RV were characterized by a Thermo Scientific Smart iTR™ in the wavenumber range of 500–4000 nm^−1^. The WA-XRD and SA-XRD patterns were recorded on a Rigaku X-ray powder diffractometer equipped with a Co source (40 kV, Kα 0.1790 nm). A TGA (TGA/DSC Thermogravimetric Analyzer, Mettler-Toledo Inc) was utilized to measure the weight loss of the OEO, OEO-loaded SMV and SMV, which were used to calculate the OEO loading amount. In a typical procedure, ~10–15 mg of a TGA sample was placed in an alumina pan and heated from 25 °C to 900 °C at a heating rate of 2 °C/min and an air flow rate of 20 mL/min. The isothermal release of OEO from the SMV was also evaluated by TGA at a constant temperature of 60 °C.

### 4.5. In Vitro Antibacterial Tests

#### 4.5.1. Preparation of Bacterial Suspension by Growth Method

*E. Coli* (ATCC 25922) and *S. epidermidis* (ATCC 12228) were obtained from American Type Culture Collection (ATCC). *E. Coli* and *S. epidermidis* were separately cultured on Luria–Bertani (LB) agar (agar 20 g/L, tryptone 10 g/L, yeast extract 5 g/L, NaCl 10 g/L, pH 7.0) media at 37 °C for 24 h. Then, single colonies were isolated in LB broth and cultured overnight in a shaking incubator at 37 °C at 225 rpm. After the overnight culture, the bacterial suspension was mixed with 15 mL of phosphate buffer saline (PBS). The turbidity of the bacterial suspension was adjusted according to the McFarland 0.5 standard to obtain a suspension of 1 × 10^8^ CFU per mL in accordance with the clinical and laboratory standards institute (CLSI) guidelines [102]. The prepared cultures were used within 30 min.

#### 4.5.2. MIC by Agar Dilution Method

The MIC of OEO and the OEO-loaded SMV were evaluated with slight modifications to the previously reported method [77] against gram-negative (*E. coli*) and gram-positive (*S. epidermidis*) bacteria models. Briefly, the stock dispersions of the OEO and OEO-loaded SMV were separately prepared and stored. From the stock suspensions, a series of diluted suspensions of OEO and OEO plus SMV with the OEO concentrations of 0.08, 0.16, 0.32, 0.64, 1.28 and 2.56 mg/mL were prepared by mixing with 25 mL of molten agar. A blank control group was prepared with 25 mL of molten agar only. Dilutions were homogenously mixed, poured into 90 mm sterile glass Petri dishes and allowed to solidify inside a laminar flow. The bacterial suspensions prepared in step Section 4.5.1 were further diluted to 1:10 in a 96-well plate. Then 8 × 5 spots were made on the agar plates from the bacterial dilution using an 8-channel pipette. After 24 h of incubation, the antibiotic dilution with the lowest OEO concentration without any colonies was considered as MIC.

#### 4.5.3. Long Term Bacterial Inhibition Test by Single Plate Serial Dilution Method

The stock dispersions of OEO-loaded SMV plus OEO, the OEO-loaded RV and free OEO were prepared by mixing with sterile water in a series of diluted suspensions with OEO concentrations of 6.4, 8.8 and 12.8 mg/mL. Bacterial suspensions (*E. coli* and *S. epidermidis*) containing 5 × 10^8^ CFU/mL bacteria were prepared using the same procedure described in Section 4.5.1, above. 100 μL of each of the suspension was pipetted into 2 mL micro-centrifugation tubes, and then 800 μL of sterile LB broth (tryptone 10 g/L, yeast extract 5 g/L, NaCl 10 g/L, pH 7.0) was added as the growth media. Afterwards, 100 μL of the diluted OEO-loaded SMV, RV plus OEO or OEO suspensions were pipetted into the micro-centrifugation tubes containing a bacterial suspension and the growth medium. The final concentrations of OEO were 0.64, 0.88 and 1.28 mg/mL. Then, the samples were incubated in a shaking incubator at 200 rpm 37 °C for 24, 48 and 72 h. All experiments were performed in triplicate (three repetitions).

The single plate serial dilution method [78] was applied for bacterial culturing. Briefly, 90 mm sterile agar plates were divided into 6 equal sections and named by the relevant dilution factor. LB agar was poured into these plates and allowed to solidify. At each time point, 200 μL of each sample was pipetted out and 6 serial dilutions (10^−1^, 10^−2^, 10^−3^, 10^−4^, 10^−5^, and 10^−6^) were prepared in a 96-well plate using a sterile PBS. Subsequently, 20 μL of each dilution was pipetted out and spotted on the relevant section of dilution on the Petri dishes. 5 Petri dishes were spotted for each sample at each time point. All the samples were incubated overnight, and the bacterial colonies were counted. The number of colonies in the range of 6–60 CFU/mL at the lowest dilution was considered acceptable and considered for the calculations. CFU per mL of sample were calculated by the following equation. The statistical significance among the similar concentrations of three treatment groups was analyzed by the multiple comparison test.
(1)$$\mathrm{Colony}\;\mathrm{forming}\;\mathrm{units}\;\mathrm{in}\;a\;\mathrm{mL}\;=\;n\;\times \left(\frac{1000\;\mu L\;}{20\;\mu L}\right)\times 10d^{\left(n+1\right)}$$
where n = number of colonies in applied area of the sample (20 μL), and d = dilution level, yielding the countable colonies. 

In order to evaluate the long-term bacterial inhibition on the carriers, similar experiments were carried out using SMV and RV at the concentration of 16.72 mg/mL (the same vermiculite concertation as the 0.88 mg/mL OEO plus SMV or OEO plus RV groups). The GC and BC were also included as the control groups. 

#### 4.5.4. Bacteria SEM Observations

Before SEM sample processing, the *E. coli* and *S. epidermis* samples were prepared according to Section 4.5.1, and treated with either OEO-loaded SMV or free OEO. A blank control group was also prepared for comparison. The bacterial samples were then fixed with 2.5% glutaraldehyde (as the total percentage in applied volume) in 2 mL sterile micro-centrifugation tubes. As another preparation step, some round coverslips were coated with 120 µL of poly-L-lysine (1 mg/mL) directly obtained from a freezer for 5 min. Then, the coverslips were washed three times with distilled water and dried before sample processing. The poly-L-lysine coated coverslips were placed in a 9-well plate and the samples, fixed by glutaraldehyde, were applied onto the coverslips. Then, the samples were stained with 1% osmium tetroxide. The excessive liquid was removed by pipette and washed sequentially with 30, 50, 70, 90, and 100% ethanol for 5 min. The samples were kept slightly wet before the chemical drying process. Then, the samples were dried with chemical drying to preserve the morphology with hexamethyldisilane (HMDS). An HMDS-ethanol (1:1) solution was applied onto the coverslips in the wells and dried in a vacuum dryer by keeping the sample wet for 3 min. 100% HMDS was once again applied to the recovered samples, which were then once again dried in the vacuum dryer for 3 min. The above step was repeated. The leftover HMDS was pipetted out, and the coverslips were further dried in a fume hood. The dried samples were glued to the carbon tabs fixed on the SEM stubs. The SEM samples were kept in a vacuum drier and sputter-coated with platinum before SEM observation. 

## 5. Conclusions

In this work, a top-down ball milling approach was used to successfully fabricate submicron-sized vermiculite particles from Australian raw vermiculite. The particle size, thickness and crystalline domain size of the milled SMV were decreased, while the apparent density of SMV increased. The particle size of SMV was in the range of 200–1000 nm with an average size of 459 nm. By mechanical mixing, OEO can be loaded by SMV. The isothermal release test showedthat SMV could slow down OEO release and significantly improve its thermal stability. OEO-loaded SMV also demonstrated enhanced antibacterial performance in in vitro antibacterial tests against both Gram-positive and Gram-negative bacteria. The small particle size, higher density and enhanced stability of OEO are crucialfor the improved antibacterial performance. It is also demonstrated that OEO-loaded SMV can adhere to the cell membrane/wall and cause membrane/wall disintegration. This work provides essential knowledge in designing advantageous carriers for volatile actives in antimicrobial applications.

## 6. Patents

A patent is resulting from this work (A semi-wet milling strategy to fabricate ultra-small nano-clay. International Patent Application Number PCT/AU2019/050575).

## Figures and Tables

**Figure 1 antibiotics-10-01324-f001:**
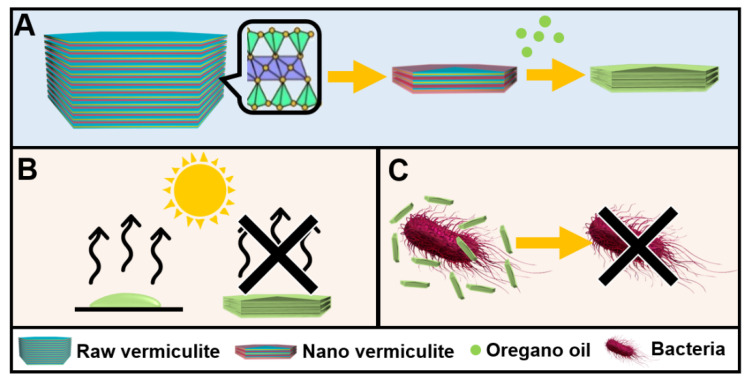
A scheme showing (**A**) the synthesis of SMV by ball milling as an OEO carrier, (**B**) the prevention of OEO loss in the isothermal release and (**C**) the long-term bacterial inhibition of OEO-loaded SMV.

**Figure 2 antibiotics-10-01324-f002:**
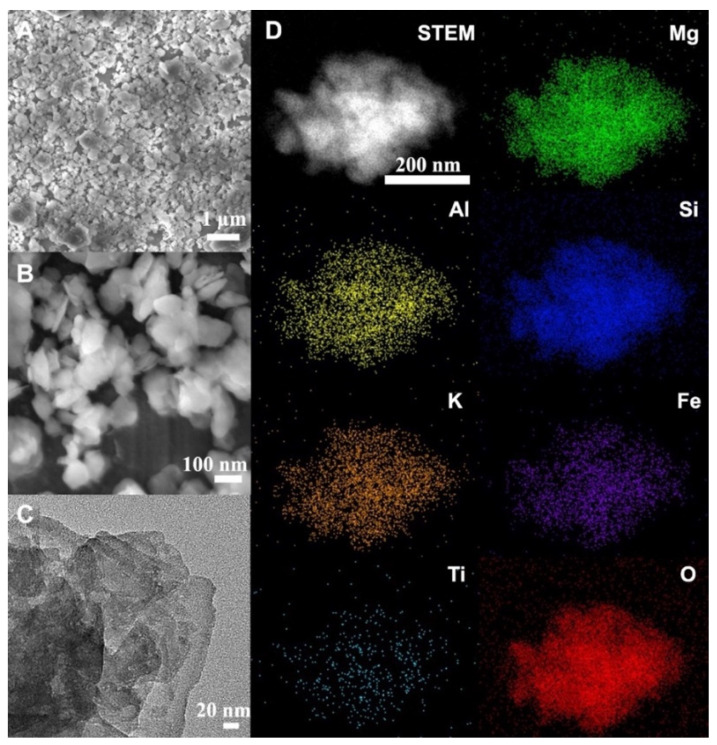
SEM (**A**,**B**), TEM (**C**) and EDS (**D**) elemental mapping images of SMV.

**Figure 3 antibiotics-10-01324-f003:**
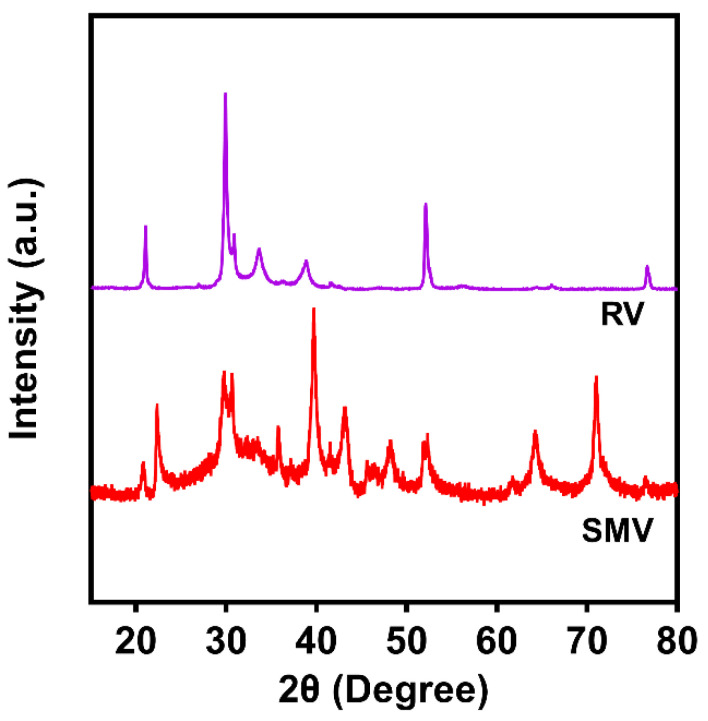
WA-XRD patterns of RV and SMV.

**Figure 4 antibiotics-10-01324-f004:**
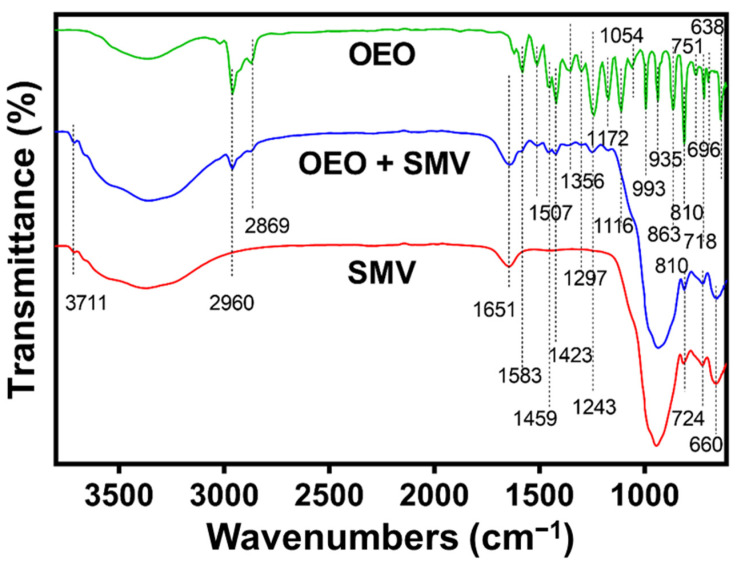
FTIR spectra of OEO, OEO-loaded SMV and SMV.

**Figure 5 antibiotics-10-01324-f005:**
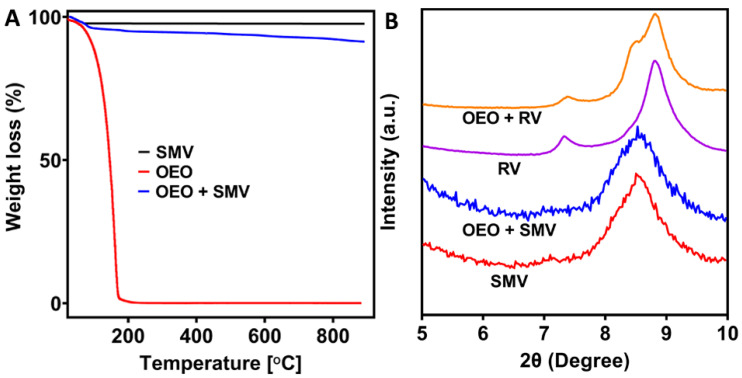
TGA (**A**) and SA-XRD patterns (**B**) of SMV before and after OEO loading.

**Figure 6 antibiotics-10-01324-f006:**
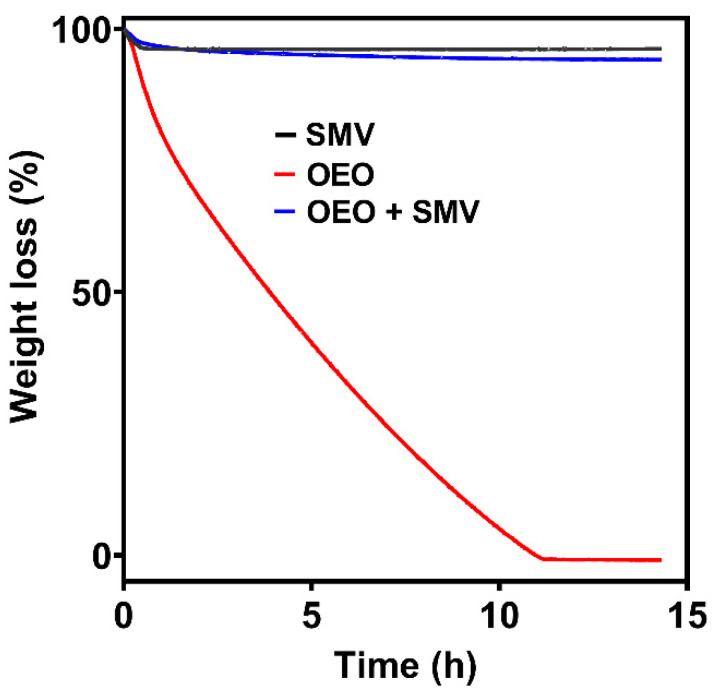
Isothermal release profiles of free OEO, SMV and OEO-loaded SMV.

**Figure 7 antibiotics-10-01324-f007:**
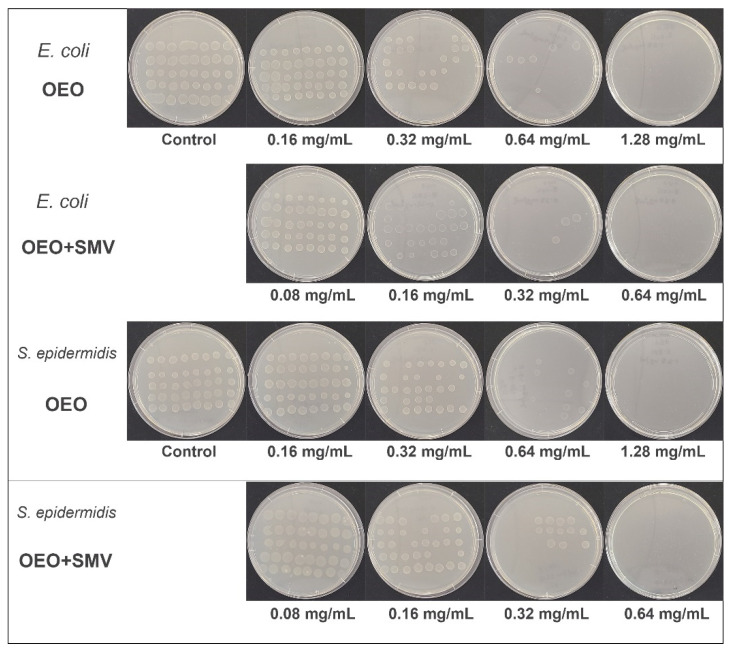
MIC tests of OEO and OEO-loaded SMV for *E. coli* and *S. epidermidis*.

**Figure 8 antibiotics-10-01324-f008:**
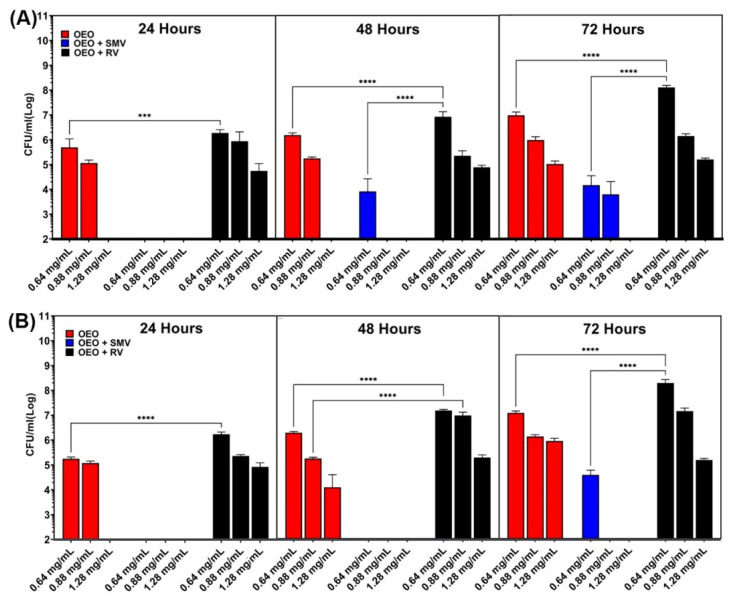
In vitro long-term bacterial inhibition efficacy of OEO loaded SMV, OEO-loaded RV and free OEO towards (**A**) *E. coli* (**B**) *S. epidermidis* (statistical significance calculated by the multiple comparison test, *** *p* = 0.0003, **** *p* < 0.0001).

**Figure 9 antibiotics-10-01324-f009:**
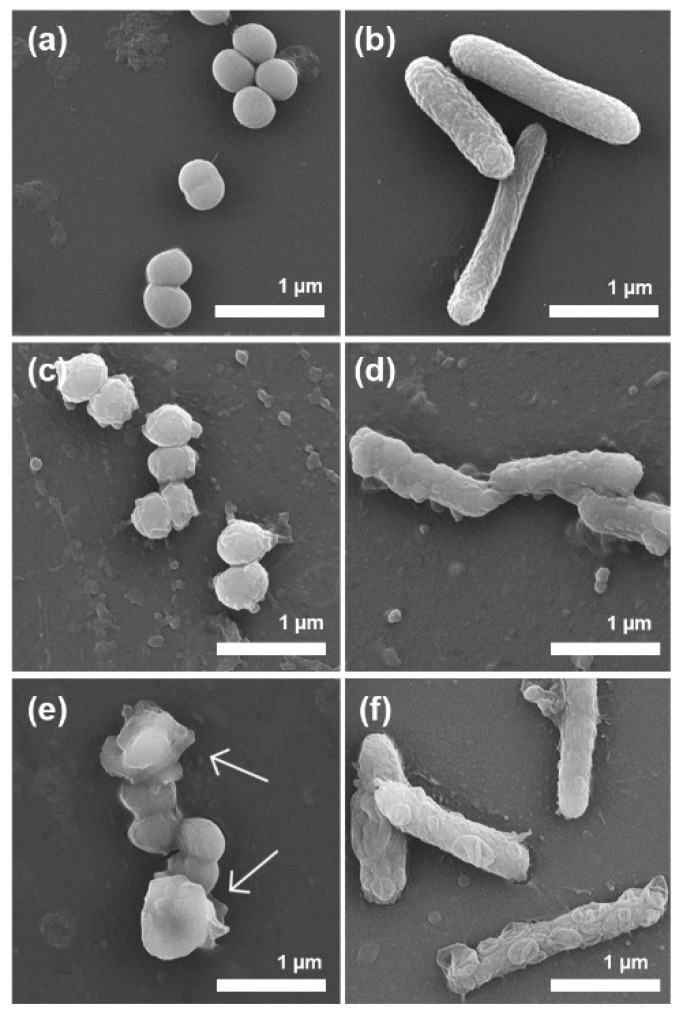
SEM images of untreated (natural) *S. epidermidis* (**a**) and *E. coli* (**b**), free OEO treated *S. epidermidis* (**c**) and *E. coli* (**d**), and OEO-loaded SMV treated *S. epidermidis* (**e**) and *E. coli* (**f**).

## Data Availability

The data presented in this study are available on request from the corresponding author. The data are not publicly available due to privacy.

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
