# Peer review of "Submicron-Sized Vermiculite Assisted Oregano Oil for Controlled Release and Long-Term Bacterial Inhibition"

_antibiotics, 2021, doi:10.3390/antibiotics10111324_

Round 1
Reviewer 1 Report
Dear Authors,
thank you for the manuscript. Before it can be recommended for publication please consider the following comments.
Introduction:
The aim should be clearly stated so that results and conclusions can be verified. Figure one could be a good graphical abstract but it should not be included at the end of the introduction.
Results and discussion
- The percentage scale is questionable to present these results. I understand that this is a percent of the control? If yes it should be included in the Figure caption.
- There is no statistical analysis in Figure 7. How did the authors compare the results?
- The authors did not provide any proof for the mechanism of action of the prepared material that should be investigated.
Materials and methods:
- Which number of the used E. coli strain?
- Celsius degrees should be written with a single symbol to avoid different versions.
- Why this particular concentration of OEO?
- What does it mean that the tests were duplicated? How many repetitions of each sample were done? There should be at least five repetitions of one sample and preferably three repetitions of the whole experiment. Otherwise the results cannot be called repeatable.
- There is no statistical analysis in the microbiological part of the manuscript.
- Why only one bacterium was used in the experiments? There is only Gram-negative model and no Gram-positive. How the authors justify that? The justification should be also included in the manuscripts. Furthermore all limitations of the studies should be also included in the text.
Conclusions
- Conclusions are not supported by the research material. Stating that this material is antibacterial based on only one strain and a single bacterium is not enough. The authors should be very specific here. Please make sure that there is a good connection between the clearly stated aim of the study and conclusions.
References: Good selection of references. I did not spot references deriving from predatory journals. Less than 20 out of 55 references is no older than five years. This could be improved.
Reviewer 2 Report
In this work, the authors reported ball-milled submicron-sized vermiculite nanoparticles as potential nano-carrier to oregano oil (OEO). The materials were characterized by Scanning and Transmission electron microscopy (TEM, SEM), Energy dispersive X-ray spectroscopy (EDS), Wide and small angle powder X-Ray Diffraction (WA-XRD, SA-XRD), FTIR and TGA. In the last part, the antimicrobial effect of OEO-loaded nano vermiculite was showed.
The work reported by Kothalawala and coworkers is interesting and could be considered for publication after minor revision.
Following are some questions/recommendations/suggestions:
- In the paper are presented submicron-sized particles. Why did the authors report that they obtained nanosized particles?
- The percentage table of elements of "nano" vermiculite from EDS should be given.
- The distribution of particle size of vermiculite should be presented.
Reviewer 3 Report
- Line 185: Please clarify if the number is 0.30% or 30%.
- Would be nice to rephrase lines 181-184. The 0% data is confusing... Is because there is no growth of bacteria? It is because of the speed of growth is reduce (and cannot be detected during the first 72 h)?
- In vitro Bacterial inhibition test: was the concentration of free OEO the same concentration as in case of encapsulated OEO? Why the authors include only one concentration in this study?
- Line 242: Would be nice if the authors express the 2 L of 10^-4 M HCl as mM concentration (concentration would be clear)
- Discussion must be improved. Now it only summarizes the results section. Please compare with other OEO-encapsulation strategies and with vermiculite encapsulations.
Round 2
Reviewer 3 Report
I appreciate the changes performed by the authors in the manuscript.